# The Mediating Effects of Affect on Associations between Impulsivity or Resilience and Internet Gaming Disorder

**DOI:** 10.3390/jcm8081102

**Published:** 2019-07-25

**Authors:** Daun Shin, A Ruem Choi, Jiyoon Lee, Sun Ju Chung, Bomi Kim, Minkyung Park, Myung Hun Jung, Dai Jin Kim, Jung-Seok Choi

**Affiliations:** 1Department of Neuropsychiatry, Seoul National University Hospital, Seoul 03080, Korea; 2Department of Psychiatry, SMG-SNU Boramae Medical Center, Seoul 07061, Korea; 3Department of Psychiatry, Hallym University Sacred Heart Hospital, Hallym University College of Medicine, Anyang 14068, Korea; 4Department of Psychiatry, Seoul St. Mary’s Hospital, The Catholic University of Korea College of Medicine, Seoul 06591, Korea; 5Department of Psychiatry and Behavioral Science, Seoul National University College of Medicine, Seoul 03080, Korea

**Keywords:** internet gaming disorder, mediation analysis, affect, impulsivity, resilience

## Abstract

Internet gaming disorder (IGD) is a new disease proposed by the Diagnostic and Statistical Manual of Mental Disorders-Fifth Edition (DSM-5), and has been studied extensively in relation to depression and impulsivity. The relationship between resilience and disease has been found in a variety of addictive disorders, but studies on IGD are lacking. In this study, 71 IGD patients and 78 healthy controls (HCs) were recruited. Impulsivity, resilience, affects, and the degree of internet game addiction were measured using formal tools. The measured values were analyzed by mediation analysis to evaluate the mediating role of affects on resilience and impulsivity related to IGD symptoms. The IGD group showed higher impulsivity, lower resilience, lower positive affect, and higher negative affect than the HC group. The mediation analysis showed that a positive affect was a mediator between impulsivity and the severity of addiction in both groups. Negative affect mediated impulsivity/resilience and the severity of addiction only in the IGD group. Although the results of this study are based on a narrow category of subjects, who are young male adults around 25 years of age, the results suggest that positive affect can be strengthened to prevent the IGD illness, and that illness symptoms may be alleviated by reducing negative affect.

## 1. Introduction

Since the development of the internet, the internet has had many impacts on our society. One of them is an internet game. Since 2005, when the phenomenon related to Internet game addiction has been reported, and various countries have received varying degrees of attention [1]. The Diagnostic and Statistical Manual of Mental Disorders-Fifth Edition (DSM-5) proposed that internet gaming disorder (IGD) is a psychiatric disorder that requires further research [2]. Additionally, the 11th revision of the International Classification of Diseases (ICD-11), which was released in 2018, includes a definition for gaming disorder [3]. The prevalence rate of IGD is 1.2–8.5%, and the prevalence rate varies according to diagnostic criteria and age of the subjects [4,5,6,7]. IGD has also been associated with psychiatric symptoms, such as mood disorders, behavior problems, and anxiety [8].

Among the various psychiatric symptoms, impulsivity has been identified as a risk factor in many addictive disorders [9]. Excessive gaming, including IGD, has been found to be associated with increased impulsivity, which has been suggested as a risk factor for conversion to disease [10,11]. Additionally, IGD leads to increased impulsivity similar to that seen with pathologic gambling and alcohol use disorders [12,13]. Furthermore, many structural and functional brain studies support increased impulsivity in patients with IGD. Patients with IGD have reduced gray matter volume in areas controlling impulsivity, such as the anterior cingulate cortex and the supplementary motor area [14]. Unlike what is observed in normal controls, the decrease in gray matter volume in IGD patients is not proportional to the degree of impulsiveness, suggesting dysfunction [15]. A study using functional magnetic resonance imaging (fMRI) reported that patients with IGD show decreased resting-state functional connectivity of the bilateral orbitofrontal cortex, which is normally associated with impulsivity [16], as well as decreased brain activity in the dorsolateral prefrontal cortex and bilateral inferior frontal gyrus [17].

Approximately 15–25% of patients with IGD are known to have depression [18,19,20]. It has also been reported that depressive symptoms improved after the IGD was cured [21]. Negative emotional states, such as depression in patients with IGD, are result of weakened functional connectivity in the default-mode-network area [22]. There is also an evidence showing that typical tactics to control emotions in the IGD group contribute to depression [23]. Resilience is one of many factors affecting emotions. Although the definition of resilience varies from one discipline to another, a common definition is the ability to be healthy, adaptive, and positively cope with change [24]. Resilience has much to do with the particular way of controlling these feelings. For example, people with high resilience use positive emotions to recover from negative feelings, such as depression [25]. There have been studies investigating how resilience affects addictive disorders by controlling positive or negative emotions. Psychological resilience has been shown to weaken the link between anxiety symptoms and addiction [20]. And internet addiction affects the relationship between psychological resilience and depressive symptoms [26]. In addition, a resting electroencephalogram (EEG) analysis study in IGD patients has suggested that resilience prevents the disease from being transmitted, as there were indirect effects of resilience on IGD through resilience-related EEG features [27].

However, the relationship between resilience, mood change, and impulsivity has not been clarified. Research on how resilience and impulsiveness affect IGD symptoms through the mediation of emotions is particularly insufficient. In this study, we investigated the difference in impulsiveness, resilience, and affect status between patients with IGD and healthy controls (HCs). We further investigate the role of affect status between impulsiveness as a risk factor and resilience as a protective factor in IGD. The hypothesis of this study is that resilience acts as a protective factor in IGD patients to reduce IGD symptoms through positive emotion, and impulsivity exacerbates IGD symptoms through negative emotions.

## 2. Materials and Methods

### 2.1. Participants

A total of 150 young male adults participated in this study, 71 of which had been diagnosed with IGD (age = 25.56 ± 5.67 years), as well as 79 healthy controls (age = 25.57 ± 4.59 years). All patients were recruited from SMG-SNU Boramae Medical Center in Seoul, South Korea, from October 2014 to September 2018. The patients with IGD were diagnosed by a clinically experienced psychiatrist according to DSM-5 criteria. None of the participants had a history of significant head injury, seizure, intellectual disability (IQ ≥ 80), or psychotic or neurological disorders. Additionally, all participants were medication-naive. Participants who spent more than 4 h per day and 30 h per week playing internet games were included in the IGD group, in order to exclude milder cases of gaming problems [28]. All participants in the HC were recruited through advertisements, none had a history of any psychiatric disorder, and all played Internet games for less than 2 h per day.

The institutional review board of the SMG-SNU Boramae Medical Center approved the study protocol (Institutional Review Board number: 16-2014-139), which adhered to the principles of the Declaration of Helsinki. All participants understood the study procedure and provided written informed consent prior to participation.

### 2.2. Measures

#### 2.2.1. Young’s Internet Addiction Test (Y-IAT)

The severity of IGD was evaluated with Young’s internet addiction test (Y-IAT). In this study, we used a scale made by modifying IAT to assess the symptoms of IGD, which consisted of 20 items, each evaluated by 5 points [28,29]. The total score ranges from 20 to 100, with higher scores reflecting a greater tendency for IGD symptoms (Cronbach’s α = 96).

#### 2.2.2. Barratt Impulsiveness Scale-11 (BIS-11)

The Barratt impulsiveness scale (BIS-11) includes 11 items rated on a four-point scale. BIS-11 assesses the degree of impulsivity, and includes three subscales: cognitive, motor, and non-planning impulsiveness (Cronbach’s α = 0.72).

#### 2.2.3. Connor-Davidson Resilience Scale (CD-RISC)

The Connor-Davidson resilience scale (CD-RISC) is a 25-item, self-reporting instrument that uses a five-point response scale. The CD-RISC total score ranges from 0 to 100, with higher scores reflecting greater resilience (Cronbach’s α = 96).

#### 2.2.4. The Positive and Negative Affect Schedule (PANAS)

The positive and negative affect schedule (PANAS) consists of a two-factor structure of 10 questions regarding 10 positive and 10 negative affect types on a five-point Likert scale (positive affect Cronbach’s α = 88; negative affect Cronbach’s α = 89).

### 2.3. Statistical Analysis

A Chi-square test and *t*-test were performed to compare the demographic and clinical characteristics of the IGD and HC groups. Pearson’s correlation analysis was conducted to examine the relationships between IGD symptoms (Y-IAT), impulsivity (BIS-11), resilience (CD-RISC), and affect (PANAS). To examine whether the quality of affect mediates the relationship between impulsivity or resilience and IGD symptoms, we performed a serial mediation analysis using the SPSS PROCESS macro, version 2.16 (model 4), developed by Hayes. In particular, we conducted the analysis using the bootstrapping method. SPSS software version 21.0 (IBM Corp., Armonk, NY, United States) was used for all data analyses. For all analyses, *p* values < 0.05 were considered to indicate statistical significance.

## 3. Results

### 3.1. Demographic and Clinical Characteristics

The mean age of all participants was 25.44 ± 5.02 years. A comparison of the demographic and clinical characteristics of the IGD and HC groups showed that the Y-IAT scores (*t*(1.706) = 8.64, *p* < 0.001), BIS-11 scores (*t*(0.001) = 5.22, *p* < 0.001), and negative scores on the PANAS (*t*(0.398) = 4.42, *p* < 0.001) were significantly higher in the IGD group than in the HC group, and the CD-RISC scores (*t*(2.972) = −5.32, *p* < 0.001) and positive scores on the PANAS (*t*(1.809) = −4.99, *p* < 0.001) were significantly lower in the IGD group than in the HC group (Table 1).

### 3.2. Correlation Analysis of Internet Gaming Disorder Symptoms, Impulsivity, Resilience, and Affect Status

In both IGD and HC groups, IGD symptoms, as measured by Y-IAT, were significantly correlated with resilience (IGD: *r* = −0.273, *p* < 0.05; HC: *r* = −0.459, *p* < 0.001), positive affect (IGD: *r* = −0.262, *p* < 0.05; HC: *r* = −0.330, *p* < 0.01), and negative affect (IGD: *r* = 0.366, *p* < 0.01; HC: *r* = 0.343, *p* < 0.01). However, there was a significant relationship between IGD symptoms and impulsivity only in the HC group (*r* = 0.262, *p* < 0.05). Impulsivity was not related to IGD symptoms in the IGD group, as shown in Table 2.

### 3.3. Mediation Analysis for Relationships between Impulsivity, Resilience, Affect Status, and IGD Symptoms in the IGD Group

As shown in Table 3, the mediating model used in this study revealed significant indirect effects of positive or negative affect on the relationship between impulsivity or resilience and IGD symptoms in the IGD group. However, the indirect effect of positive affect on the relationship between resilience and IGD symptoms was not significant. The confidence intervals for the indirect effect of affect contained 0, suggesting that the positive affect was not a significant mediator in the effect of resilience on IGD symptoms. Furthermore, there were no significant direct effects in any of these models (Figure 1). To summarize these results, the level of positive or negative affect mediated the relationship of impulsivity and increased IGD symptoms in patients with IGD. Specifically, resilience had a negative correlation with the level of negative affect, which was strongly associated with IGD symptoms in the IGD group (Table 3).

### 3.4. Mediation Analysis for Relationships between Impulsivity, Resilience, Affect, and IGD Symptoms in the HC Group

As shown in Table 4, the mediating model used in this study revealed only a significant indirect effect of positive affect on the relationship between impulsivity and IGD symptoms in the HC group. In contrast to the results in the IGD group, there were significant direct effects of positive or negative affect on the relationship between resilience and IGD symptoms in the HC group (Figure 2 and Table 4).

## 4. Discussion

In this study, the IGD group showed higher impulsivity, lower resilience, lower positive emotions, and higher negative emotions than the HC group. A correlation analysis revealed that IGD symptoms were correlated with resilience, negative affect, or positive affect in both the IGD and HC groups. No correlation was detected between impulsivity and IGD symptoms in the IGD group, as opposed to the HC group, in which a significant correlation was detected. This association was analyzed by mediation analysis; the direct effects of resilience and IGD symptom severity were detected in the HC group, but not in IGD patients. In the IGD group, both negative and positive emotions mediated the relationship between impulsivity and addiction. The relationship between resilience and addiction was significantly mediated only by negative emotion. On the other hand, only positive emotions mediated the relationship between impulsivity and addiction in HC group. All of these results were obtained in young male adults.

Positive affect served as a protective mediator between impulsivity and IGD symptoms in both groups. Positive affect plays a role in depression, and also affects the quality of life for cancer survivors, as well as the prognosis after lumbar surgery [30,31,32]. Positive emotions also affect various cognitive functions, such as decision-making. Those who often feel positive affirmations gain more compensation through more risky investments in decision-making tasks, and feel more positive emotions at the end of the assignment. This means that a person with a positive affect will use a variety of resources to achieve better results [33]. In a cue craving test in patients with nicotine addiction, the cue that showed positive emotions resulted in stronger cravings, indicating the effect of positive emotions on addiction [34]. As dopamine affects both positive and rewarding cognition processes, the “dopamine hypothesis of positive affect” was raised in a 2014 review article [35]. In addition, the more positive affect traits there are, the more goal-directed information is well maintained in work memory [36]. Consistent with these results, in this study we confirmed that the higher the tendency of positive affect, the lower the degree of addiction by suppressing impulsivity. This means that the higher tendency for positive affect, the more likely it is that the process of working memory will change, and the most efficient choice will be made based on various alternative strategies. Therefore, we confirmed the possibility of preventing mental illness by strengthening positive affect in people vulnerable to addiction.

The differences in the direct effects of resilience between the HC and the IGD groups suggest that resilience directly protects against IGD symptoms in the HC group, but only has indirect effects on addiction symptoms through negative affect in the IGD group. In the disease condition, it may not be possible to see the direct effect of resilience, because there is no alternative strategy seen in it, as cognitive impairment arises after the disease sets in. Cognitive deficits are widely found in many addiction disorders, including IGD [37,38]. Especially in the IGD group, loss is more frequent than in normal controls, by making disadvantageous and dangerous choices in decision-making [39]. In addition, an fMRI study in patients with IGD showed a decrease in function due to hyperactivity of brain regions associated with attention, working memory, and cognitive control during gaming [40]. This finding is supported by the result of another fMRI study, in which activity in the right dorsolateral prefrontal cortex, which regulates decision-making, was reduced in the IGD group, leading to more risky decisions [41]. The tendency to make risky decisions is consistent with studies in IGD males [42]. This risky decision-making may be the result of reduced compensation sensitivity and impulse control capability in the IGD group [43]. To find a suitable alternative strategy, it is necessary to not make a hasty decision, but rather to consider the long-term benefits rather than the immediate benefits, an ability that is disturbed by various cognitive deficiencies. One study observed a change in risk avoidance after a stressful situation, confirming the effect of stress on cognition [44]. Also, past stress affects memory, learning, and problem-solving, and as a result, alternative strategies are formed to respond to current stress [45]. Thus, in the IGD group in our study, resilience mediated overwhelming negative emotions and cognitive errors to help cope with the disease, but did not directly affect disease severity.

Cognitive impairment is affected not only by disease but also by negative affect. Many studies have shown a relationship between cognitive errors and negative emotions, such as depression, and the cognitive behavioral therapy for treating negative emotions has also been recognized as effective [46,47,48]. In addition, the relationship between cognitive deficit and negative affect, such as depression, has been confirmed [49]. Patients with depression are unable to appropriately appreciate the future because they have rigid thoughts, which has been shown to affect emotional acceptance [48,50]. Anxiety, another negative affect, is also strongly associated with cognition. In a 2012 study, patients with anxiety identified a tendency toward lower risk-taking when making a decision [51,52]. In addition, the function of conceptual cognition decreases in patients with anxiety disorders [53]. These cognitive errors are the reasons for reduced task performance in patients with anxiety [54]. This context can also explain why negative affect significantly mediated the disease severity in the IGD group but not in the HC group.

The final conclusions of this study suggest that IGD can be prevented by enhancing positive affect, and that addiction can be moderated by modulating negative affect after IGD. Indeed, previous studies have suggested that strengthening active coping among the various aspects of resilience may provide a key benefit in patients with psychiatric disorders [55]. There are also previous results that indicate that improving active coping to overcome anxiety improves the prognosis, even when there is a physical disease [56,57]. However, there is still a lack of research on which coping skills are effective. Through this study, we propose that the skills that strengthen positive emotions before the illness are more effective, and skills that weaken negative emotions after the disease can be more effective coping skills for IGD.

However, there are limitations to consider when interpreting our results. First, as this study was conducted in young male adults, it is not yet clear whether these results are consistent across all age groups or female groups with IGD. Many IGD studies have been conducted in adolescents, and school-age mental illnesses differ from adult-onset mental illnesses in clinical manifestations. A difference has been detected between the appetitive aspect and the compulsive aspect in adolescents and adults with addiction disease [58]. In addition, gender differences in the severity of addiction and motivation for playing games were confirmed in patients with a game addiction in a previous report [59]. Therefore, it is necessary to confirm similar results in patients of various sexes and ages before introducing the results of the present study into comprehensive medical practice. Second, a mediation analysis was used to investigate the causality among resilience, impulsivity, and affect, but it is necessary to question it. Although the direction of etiology can be suggested by confirming the mediation effect, all factors were measured cross-sectionally, not taking into account temporal effects. Although the etiology of IGD is still under investigation, the current analysis assumed that impulsivity and resilience are characteristics of individuals that do not change significantly, and that they are controlled by emotions. However, as an affect may be influenced as a result of psychological stress, due to high impulsiveness and low resilience, it may be necessary to verify the inverse relationship of the causality identified in this study in the future [60]. In particular, studies are needed to clarify causality, including temporal effects of those variables. Third, there were no direct effects of impulsivity or resilience on IGD symptoms. In this mediation analysis, this may not be a problem. This is because this study used the bootstrap test of the indirect effect, which is a way to analyze the mediation effect even if there is not enough of a direct effect [61].

## 5. Conclusions

In this study, the HC group showed a direct correlation between the degree of resilience and the severity of IGD symptoms, and only positive emotions mediated impulsivity and the severity of IGD symptoms. In the IGD group, the effects of impulsivity and resilience on the severity of addiction were mediated by negative affect, with no direct effects of resilience on the IGD. These results suggest that controlling emotions could help prevent disease and relieve symptoms. To confirm the results of the present study, future research should examine the effects of changes in emotions while taking drugs. The results also suggest that identifying the mediation effects of various factors associated with game addiction may be helpful in understanding the psychological etiology of a disease.

## Figures and Tables

**Figure 1 jcm-08-01102-f001:**
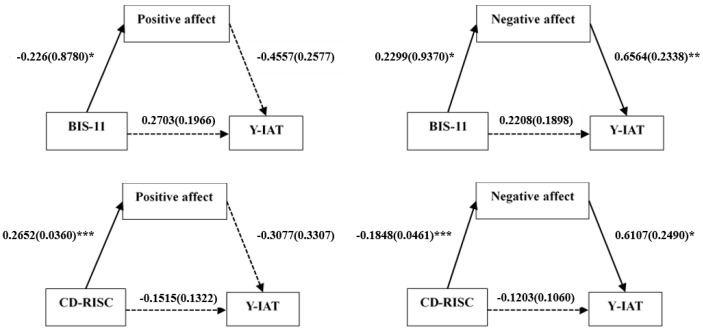
Mediation effects of affect status on the relationship between impulsivity or resilience and the severity of internet gaming disorder (IGD) symptoms in the patient group. Unstandardized coefficients are presented with standard errors in parentheses. * *p* < 0.05, ** *p* < 0.01, *** *p* < 0.001. Y-IAT: Young’s internet addiction test; BIS-11: Barratt impulsiveness scale-11; CD-RISC: Connor-Davidson resilience scale.

**Figure 2 jcm-08-01102-f002:**
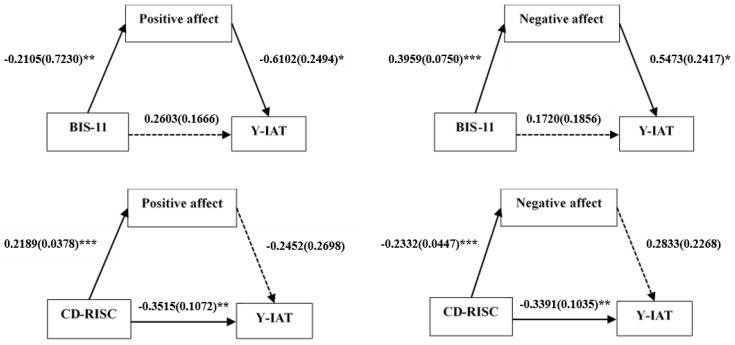
Mediation effects of affect status on the relationship between impulsivity or resilience and the severity of internet gaming disorder symptoms in the healthy control (HC) group. Unstandardized coefficients are presented with standard errors in parentheses. * *p* < 0.05, ** *p* < 0.01, *** *p* < 0.001. Abbreviations: Y-IAT: Young’s internet addiction test; BIS-11: Barratt impulsiveness scale-11; CD-RISC: Connor-Davidson resilience scale.

**Table 1 jcm-08-01102-t001:** Demographic and clinical characteristics.

Variables	IGD (*n* = 71), M ± SD	HC (*n* = 79), M ± SD	*t*	*p*-Value
Age	25.56 ± 5.669	25.57 ± 4.593	−0.007	0.994
Y-IAT	57.45 ± 15.979	35.89 ± 14.587	8.641 ***	<0.001
BIS-11	66.35 ± 9.811	57.96 ± 9.830	5.224 ***	<0.001
CD-RISC	50.79 ± 18.723	66.11 ± 16.534	−5.324 ***	<0.001
PANAS-Positive	24.55 ± 7.485	30.28 ± 6.567	−4.993 ***	<0.001
PANAS-Negative	27.32 ± 7.966	21.72 ± 7.546	4.422 ***	<0.001

*** *p* < 0.001. Correlation among variables in the IGD and HC groups Abbreviations: IGD: Internet gaming disorder; *n*: number; HC: healthy controls; M: mean; SD: standard deviation; Y-IAT: Young’s internet addiction test; BIS-11: Barratt impulsiveness scale-11; CD-RISC: Connor-Davidson resilience scale; PANAS: the positive and negative affect schedule.

**Table 2 jcm-08-01102-t002:** Correlations between IGD symptoms, impulsivity, resilience, and affect status.

**IGD (*n* = 71)**	**Y-IAT**	**BIS-11**	**CD-RISC**	**PANAS-Positive**	**PANAS-Negative**
Y-IAT	1				
BIS-11	0.228	1			
CD-RISC	−0.273 *	−0.442 ***	1		
PANAS-Positive	−0.262 *	−0.292 *	0.663 ***	1	
PANAS-Negative	0.366 **	0.283	−0.434 ***	−0.055	1
**HC (*n* = 79)**	**Y-IAT**	**BIS-11**	**CD-RISC**	**PANAS-Positive**	**PANAS-Negative**
Y-IAT	1				
BIS-11	0.262 *	1			
CD-RISC	−0.459 ***	−0.601 ***	1		
PANAS-Positive	−0.330 **	−0.315 **	0.551 ***	1	
PANAS-Negative	0.343 **	0.516 ***	−0.511 ***	−0.200	1

* *p* < 0.05, ** *p* < 0.01, *** *p* < 0.001. Abbreviations: IGD: internet gaming disorder; *n*: number; HC: healthy control. Y-IAT: Young’s internet addiction test; BIS-11: Barratt impulsiveness scale-11, CD-RISC: Connor-Davidson resilience scale; PANAS: the positive and negative affect schedule.

**Table 3 jcm-08-01102-t003:** Model of mediation analysis of affect status between impulsivity or resilience and severity of IGD symptoms in the IGD group.

Paths	Boot Indirect Effect	Boot SE	LLCI	ULCI
Impulsivity → positive affect → IGD	0.1014	0.0737	0.0014	0.3088
Impulsivity → negative affect → IGD	0.1509	0.0790	0.0317	0.3505
Resilience → positive affect → IGD	−0.0816	0.0804	−0.2753	0.0530
Resilience → negative affect → IGD	−0.1129	0.0519	−0.2427	−0.0296

Abbreviations: IGD: internet gaming disorder; Boot SE: Bootstrap for Standard Error; CI: confidence interval: LLCI: lower-level confidence interval; ULC: upper-level confidence interval.

**Table 4 jcm-08-01102-t004:** Model of mediation analysis of affect status between impulsivity or resilience and severity of IGD symptoms in the HC group.

Paths	Boot Indirect Effect	Boot SE	LLCI	ULCI
Impulsivity → positive affect → IGD	0.1284	0.0803	0.0154	0.3351
Impulsivity → negative affect → IGD	0.2167	0.1333	−0.0112	0.5268
Resilience → positive affect → IGD	−0.0537	0.0721	−0.1991	0.0835
Resilience → negative affect → IGD	−0.0661	0.0649	−0.2084	0.0457

Abbreviations: IGD: internet gaming disorder; HC: healthy control; Boot SE: Bootstrap for Standard Error; CI: confidence interval: LLCI: lower-level confidence interval; ULCI: upper-level confidence interval.

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
