# Peer review of "The Mediating Effects of Affect on Associations between Impulsivity or Resilience and Internet Gaming Disorder"

_jcm, 2019, doi:10.3390/jcm8081102_

Round 1
Reviewer 1 Report
Dear authors,
I have read with interest your manuscript which is well written and organized. I recoomand to change the title and part of your introduction of your work, since the measure IAT assesses Internet Addiction test and not Internet Gaming Disorder. It is not adequate to speak about IGD when the instrument used in this manuscript evaluates the internet addiction and in general technological addiction. I warmly reccomand to replace Internet gaming disorder with technological addiction or better internet addiction. in this case a modified introduction focused on technological addictions should be provided.
Finally, a brief discussion of these results obtained in male group in comparison with those reported in literature should be added.
Author Response
We highly appreciate your critical review that could be helpful to improve our paper. As you commented, we realized that our previous version of the manuscript had shortcomings and we tried to elaborate the manuscript throughout the paper. Detailed contents revised are as below (also marked in the revised manuscript using track-change function in MS Word).
Reviewer #1
Dear authors,
1. I have read with interest your manuscript which is well written and organized. I recommend to change the title and part of your introduction of your work, since the measure IAT assesses Internet Addiction test and not Internet Gaming Disorder. It is not adequate to speak about IGD when the instrument used in this manuscript evaluates the internet addiction and in general technological addiction. I warmly recommend to replace Internet gaming disorder with technological addiction or better internet addiction. in this case a modified introduction focused on technological addictions should be provided.
>>> Response 1: Thank you for your comments. As described in the subject recruitment of the method section, we recruited subjects who were diagnosed with the internet gaming disorder according to the DSM-5 diagnostic criteria. So it would be more accurate to express them as internet gaming disorder than internet addiction.
In case of IAT scale, we used modified version of IAT scale to evaluate the severity of internet gaming disorder. This was cited in the previous reports (Son, K.L., Choi, J-S., Lee, J., Park, S.M., Lim, J.A., Lee, J.Y., . . . Kwon, J.S., Neurophysiological features of Internet gaming disorder and alcohol use disorder: a resting-state EEG study. Translational psychiatry 2015, 5(9), p. e628.; Park, M., Choi, J-S., Park, SM., Lee, J-Y, Jung, H.Y., Sohn, B.K., Kim, S.N., and Kwon, J.S., Dysfunctional information processing during an auditory event-related potential task in individuals with Internet gaming disorder. Translational Psychiatry 2016,6, p. e721.)
Therefore, we made the contents clearer in the revised manuscript as follows:
→ 2.2.1. Young’s Internet Addiction Test (Y-IAT)
The severity of IGD was evaluated with Y-IAT. In this study, we used a scale made by modifying IAT to assess the symptoms of IGD, and it consists of 20 items which are evaluated by 5 points [28, 29]. The total score ranges from 20 to 100, with higher scores reflecting a greater tendency for IGD symptoms (Cronbach’s α = .96). (in the measures, page3 , line 96~97)
2. Finally, a brief discussion of these results obtained in male group in comparison with those reported in literature should be added.
>>> Response 2: We appreciate your comment. We added sentences about male subjects in the present study in the discussion of the revised manuscript as follows:
-à “All these results were obtained in young male adults.” (in the discussion, page 6, line 194~195 )
“This finding is supported by the result of another fMRI study, in which activity in the right dorsolateral prefrontal cortex, which regulates decision making, was reduced in the IGD group, leading to more risky decisions [41]. The tendency to make this risky decision is consistent with studies in IGD males [42].” (in the discussion, page7, line 220~221)
Reviewer 2 Report
Review for manuscript ID # jcm-552809
Brief Summary
This study evaluated 71 participants diagnosed with Internet Gaming Disorder (IGD) with regards to IGD symptomology, impulsivity, resilience, and positive and negative affect, and compared them to healthy controls of the same age (n = 79). Statistical analyses compared average levels of these constructs, correlations among them, and assessed eight mediation models investigating if positive or negative affect mediated the relationships between either impulsivity or resilience and IGD symptomology. Results indicated several significant indirect effects and differences between study groups in hypothesized directions.
Broad Comments
The authors conducted a methodologically rigorous study evaluating a relevant and emerging public health concern, Internet Gaming Disorder. There are many strengths of this study. For example, the statistical results are clearly presented, and mediation was assessed using the product of coefficients method. Please see below some suggestions that I offer in hopes that they may improve the manuscript:
Specific Comments
1. There is a typo in Tables 3 and 4 in the “Paths” column. It only says “positive affect” but two of the rows should say “negative affect”
2. There is a typo in Table 4 in the last row. The same value (-.2084) is shown for both the lower and upper limits of the confidence interval.
3. Lines 263 to 275 seem out of place. Perhaps they are there by mistake and should be deleted?
4. I had difficulty understanding what was meant by the sentence in lines 67 and 68. How did the resting EEG analysis study suggest that resilience prevents the disease from being transmitted? Perhaps including an additional sentence or two clarifying how this link was found would help clarify this study for the reader.
5. Although the authors suggest that causality should be identified in a future study in Lines 260 – 262, the language used to interpret the mediation results throughout the Results and Discussion sections often suggests causality. This is problematic, particularly since at a minimum the temporal ordering of the predictor, mediator, and outcome is needed to establish causality, and this study assessed all three at the same time. Perhaps including as a limitation of this study in the Discussion section that all variables were assessed cross-sectionally instead of longitudinally, and that more research is needed to establish causality, would remind the reader to not over-interpret the results?
6. Similarly, in the Results section on lines 153 and 154 it is suggested that resilience decreased the level of negative affect, when the actual statistical analysis only evaluated whether resilience was statistically associated with lower levels of negative affect. Perhaps the word decreased could be changed to one that does not imply causality?
7. The Baron & Kenny (1986) method of evaluating mediation requires that there be a direct effect between the predictor and outcome. Although more recent literature on mediation argues that this is not the case (e.g., Zhao, Lynch, & Chen, 2010) some readers may be concerned that a direct effect was not found for some of the mediation models. Perhaps the authors could add a sentence or two in the Discussion section about how the lack of a direct effect is not problematic?
Author Response
We highly appreciate your critical review that could be helpful to improve our paper. As you commented, we realized that our previous version of the manuscript had shortcomings and we tried to elaborate the manuscript throughout the paper. Detailed contents revised are as below (also marked in the revised manuscript using track-change function in MS Word).
Reviewer #2
Brief Summary
This study evaluated 71 participants diagnosed with Internet Gaming Disorder (IGD) with regards to IGD symptomology, impulsivity, resilience, and positive and negative affect, and compared them to healthy controls of the same age (n = 79). Statistical analyses compared average levels of these constructs, correlations among them, and assessed eight mediation models investigating if positive or negative affect mediated the relationships between either impulsivity or resilience and IGD symptomology. Results indicated several significant indirect effects and differences between study groups in hypothesized directions.
Broad Comments
The authors conducted a methodologically rigorous study evaluating a relevant and emerging public health concern, Internet Gaming Disorder. There are many strengths of this study. For example, the statistical results are clearly presented, and mediation was assessed using the product of coefficients method. Please see below some suggestions that I offer in hopes that they may improve the manuscript:
Specific Comments
1. There is a typo in Tables 3 and 4 in the “Paths” column. It only says “positive affect” but two of the rows should say “negative affect”
>>> Response 1: Thank you for your comments. This is an error and my mistake. I am really grateful to you for discovering these mistakes and checking it very carefully. I am sorry that my mistake is obvious.
We changed table 3 in the revised manuscript as below.
→
Table 3. Model of mediation analysis of affect status between impulsivity or resilience and severity of IGD symptoms in the IGD group.
Paths | Boot Indirect Effect | Boot SE | LLCI | ULCI |
Impulsivity → positive affect → IGD Resilience → positive affect → IGD Resilience → negative affect → IGD | .1014 .1509 -.0816 -.1129 | .0737 .0790 .0804 .0519 | .0014 .0317 -.2753 -.2427 | .3088 .3505 .0530 -.0296 |
Abbreviations: IGD: Internet gaming disorder, CI: Confidence interval, LLCI: lower level confidence interval, ULCI : upper level confidence interval.
(in the Table 3, page 5)
2. There is a typo in Table 4 in the last row. The same value (-.2084) is shown for both the lower and upper limits of the confidence interval.
>>> Response 2: Thank you for your corrections. We modified table 4 as follows.
→
Table 4. Model of mediation analysis of affect status between impulsivity or resilience and severity of IGD symptoms in the HC group.
Paths | Boot Indirect Effect | Boot SE | LLCI | ULCI |
Impulsivity → positive affect → IGD Impulsivity → negative affect → IGD Resilience → positive affect → IGD Resilience → negative affect → IGD | .1284 .2167 -.0537 -.0661 | .0803 .1333 .0721 .0649 | .0154 -.0112 -.1991 -.2084 | .3351 .5268 .0835 .0457 |
Abbreviations: IGD: Internet gaming disorder, HC: healthy controls, CI : Confidence interval, LLCI : lower level confidence interval, ULCI : upper level confidence interval.
(in the Table 4, page 6)
3. Lines 263 to 275 seem out of place. Perhaps they are there by mistake and should be deleted?
>>> Response 3: Thank you for kindly giving us this information. It is really my mistake. This part was deleted.
4. I had difficulty understanding what was meant by the sentence in lines 67 and 68. How did the resting EEG analysis study suggest that resilience prevents the disease from being transmitted? Perhaps including an additional sentence or two clarifying how this link was found would help clarify this study for the reader.
>>> Response 4: Thank you for your comments. It seems to have caused confusion to the readers when explaining the contents too briefly. A description of the study method was added.
→ In addition, a resting EEG analysis study in IGD patients suggested that resilience prevents the disease from being transmitted, in which there were indirect effects of resilience on IGD through resilience-related EEG features [27]. (in the Introduction, page 2, 68~69.)
5. Although the authors suggest that causality should be identified in a future study in Lines 260 – 262, the language used to interpret the mediation results throughout the Results and Discussion sections often suggests causality. This is problematic, particularly since at a minimum the temporal ordering of the predictor, mediator, and outcome is needed to establish causality, and this study assessed all three at the same time. Perhaps including as a limitation of this study in the Discussion section that all variables were assessed cross-sectionally instead of longitudinally, and that more research is needed to establish causality, would remind the reader to not over-interpret the results?
>>> Response 5: Thank you for your critical comments. As readers may interpret the results in an exaggerated way, your comments will be accepted and we added it to the limitation in the revised manuscript as follows.
→ Secondly, a mediation analysis was used to investigate the causality among resilience, impulsivity, and affect, but it is necessary to question it. Although the direction of etiology can be suggested by confirming the mediation effect, all factors were measured cross-sectionally, not taking into account temporal effects. Although the etiology of IGD is still under investigation, the current analysis assumed that impulsivity and resilience are characteristics of individuals that do not change significantly, and that they are controlled by emotions. However, as affect may be influenced as a result of psychological stress due to high impulsiveness and low resilience, it may be necessary to verify the inverse relationship of the causality identified in this study in the future [60]. In particular, studies are needed to clarify causality, including temporal effects of those variables. (in the Discussion, page 7~8, 265~272.)
6. Similarly, in the Results section on lines 153 and 154 it is suggested that resilience decreased the level of negative affect, when the actual statistical analysis only evaluated whether resilience was statistically associated with lower levels of negative affect. Perhaps the word decreased could be changed to one that does not imply causality?
>>> Response 6: Thank you for your comments. In response to your comments, we have modified the use of words to suggest causality as follows.
→ To summarize these results, the level of positive or negative affect mediated the relationship of impulsivity and increased IGD symptoms in patients with IGD. Specifically, resilience had negative correlation with the level of negative affect, which was strongly associated with IGD symptoms in the IGD group (Table 3). (in the Result 3.3., page 4, 154~158.)
7. The Baron & Kenny (1986) method of evaluating mediation requires that there be a direct effect between the predictor and outcome. Although more recent literature on mediation argues that this is not the case (e.g., Zhao, Lynch, & Chen, 2010) some readers may be concerned that a direct effect was not found for some of the mediation models. Perhaps the authors could add a sentence or two in the Discussion section about how the lack of a direct effect is not problematic?
>>> Response 7: Thank you for your comments. As your suggestions, we added some sentences in the limitation of the discussion section in the revised manuscript as follows.
→ Third, there were no direct effects of impulsivity or resilience on IGD symptoms. In this mediation analysis, this may not be a problem. This is because this study used the bootstrap test of the indirect effect, which is a way to analyze the mediation effect even if there is not enough direct effect [61].
(in the Discussion, page 6, 194~197.)